# The Meaning of a Group Facilitation Training to Creative Arts Therapists Working in the Public Education System

**DOI:** 10.3390/children10060933

**Published:** 2023-05-25

**Authors:** Efrat Roginsky, Atara Ben-Haim, Talya Cooper, Shoval Ben-Simon, Dafna Regev, Sharon Snir

**Affiliations:** 1School of Creative Arts Therapies, University of Haifa, Haifa 3498838, Israel; roginskyefrat@gmail.com (E.R.);; 2The Emili Sagol Creative Arts Therapies Research Center, University of Haifa, Haifa 3498838, Israel; 3The Interdisciplinary Research Center for Arts and Spirituality: Therapy, Education and Society, Tel Hai College, Tel Hai 1220800, Israel; sharonsnir@gmail.com; 4Art Therapy MA Program, Tel Hai College, Tel Hai 1220800, Israel

**Keywords:** creative arts therapies, education system, group facilitation

## Abstract

Many creative arts therapists who provide group therapy to children and adolescents in the Israeli education system do not feel they were sufficiently trained as group facilitators. Group facilitation training was provided over the course of two consecutive years by a regional support center to over 40 creative arts therapists and their clinical supervisors working in the Israel Ministry of Education. A two-stage qualitative research project examined the participants’ experiences during this training. Interviews were conducted regarding the therapists’ first-year experiences. A questionnaire was administered at the end of the second year. Both were analyzed according to the Consensual Qualitative Research method. The research findings pertain to the participants’ perceptions of group arts therapy in the Israeli education system and included the development of unique group models, the advantages and power of group therapy at school, and the intimidating and disruptive experiences of school settings. The participants also provided their impressions of the training course: their growing confidence and skills, and the many changes required in group arts therapy at public schools to provide more professional and efficient service. The discussion centers on the value of group arts therapy in the education system and the steps needed to enhance therapists’ confidence and efficiency in this field.

## 1. Introduction 

In recent years, there is a growing use of creative arts therapies in the educational system. Sometimes, due to a shortage of resources, school children receive therapy in groups; unfortunately, therapists facilitating these groups can encounter challenging working conditions and lack adequate training. This paper describes two-stage research, which examined the perceptions of creative arts therapists who participated in a unique group facilitation training at a regional support center (RSC) in central Israel.

### 1.1. Group Arts Therapy with Children and Adolescents in the Israeli Education System

Group therapy offers children various advantages: developing self-awareness through an ongoing, facilitated dialogue with their peers, having a safe space to meet their social challenges with more adaptive methods [1,2], and acquiring new social skills with the ongoing influence of their groupmates’ communications and behavior [3]. Numerous studies have shown that children use the arts more naturally to express their inner worlds and the complexities of life [4,5,6]. Therefore, the theory suggests that children use arts-based interventions naturally to share personal topics and inner conflicts they face during their everyday lives [2,7,8,9,10].

Group arts therapy can take place in a wide variety of settings, with children and adolescents. The literature has shown the positive contribution of the arts to group therapy with this population [11]. Preliminary qualitative studies support this notion; these studies describe, for instance, the benefits of group arts therapy for coping with schoolyard bullying [12], empowering children with a sibling with a disability [13], and developing the social skills of individuals on the autism spectrum [14]. Such groups are frequently chosen by many adolescents, who find them safe and supportive enough to allow for connection and communication with their peers [15,16]. These groups increase the adolescents’ mental and emotional adaptation [17] and address depression and anxiety [18].

In recent decades, the Israeli education system hired creative arts therapists, who offered psychotherapy to public-school children in need [19] and made therapy more accessible to everyone [20,21] and less stigmatizing than other types of counseling within the community [21]. Creative arts therapy in public schools is part of Israeli social integration policy and law. The Ministry of Education regularly funds the integration of children with special needs in public schools and kindergartens. Close to 10 thousand health professionals, including 3600 creative arts therapists, work in the Israeli school system. They see clients in their natural environment and collaborate with the multi-professional school staff [6,22]. Due to budgetary limitations and the wish to provide services to as many students as possible [21,23], creative arts therapists are expected to provide group therapy.

### 1.2. The Background to Current Study: A Group Facilitation Training

In 2017, the Hefer-Sharon Regional Support Center (RSC) in central Israel, one of 68 other RSC branches in the country, offered a group facilitation course to creative arts therapists employed at this RSC. Sixty supervisors and creative arts therapists who worked at this RSC at the time with primary and middle schools and kindergartens enrolled in this theoretical and practical course. The course was based on the concept of “the group as a whole”. According to this approach, people are born and develop within a social context; therefore, therapy in groups can be even more efficient than individual treatment [24,25]. The training included 60 h per year for two successive years; it was optional for creative arts therapists and mandatory for their supervisors. The program sought to help participants gain a profound theoretical understanding of the subject, acquire knowledge of possible applications that are more relevant to children at school, and enhance their motivation to explore even more perspectives and techniques of group facilitation. Two professional groups participated in the first year of the course: seven RSC supervisors underwent intensive instruction on group work with children and group supervision provided by an experienced clinical psychologist. A second group of creative arts therapists had a weekly supervision session regarding their groups of children, with a monthly session on group theory.

During the second year, the supervisors’ class continued their training in the same way, as more therapists joined, and they trained under two experienced group facilitators—a social psychologist and a psychodrama therapist. They experienced dynamic group sessions, theory lessons, and guest workshops in various art modalities.

The current study was part of an integrated intervention that the RSC initiated: training followed by research aiming to enhance systemic change. This occurred while assessing this process as a whole: the training participants’ changing views and experiences regarding group arts therapy in educational settings, as well as the meaning of training during the two years.

## 2. Materials and Methods

### 2.1. Participants

The study followed the group training in two sequential stages: during and after its 1st year and after its 2nd year. Four supervisors and seven creative arts therapists out of the 20 course participants who were contacted agreed to be interviewed during the first year of this study. They were all women, creative arts therapists, and clinical supervisors working in various public schools and special education kindergartens. Each of them facilitated approximately 2–12 (M = 5.7, SD = 3.44) therapy groups per year. These therapists had at least eight years of experience in their profession. During the second year, all the 34 creative arts therapists—the course participants (not the supervisors) were invited to participate in the study (the data collection was anonymous during the second stage of the study. Hence, the total number of participants cannot be specified). The participants were all women in their 30–50s, with 15 or more years of experience in the profession and the education system. They were working in schools, special education kindergartens, and early childhood care centers. Most participants worked in the Hefer-Sharon RSC, but a few were from neighboring RSCs. The inclusion criterion was enrollment in the training course during that specific year.

### 2.2. The Research Tools

#### 2.2.1. Semi-Structured In-Depth Interview

Semi-structured in-depth interviews were conducted during the first year of the research at a convenient time and place of their choice. These interviews had two aims: to learn about participants’ perceptions of group arts therapy in the education system and to assess the impact of their training. The first-year participants were interviewed twice: toward the end of this year and a few months later. The same interviews were conducted with everyone: creative arts therapists and supervisors; in addition, the RSC supervisors were asked about their impressions of the therapist supervision groups that they facilitated during this year. Each interview began with a few introductory questions, followed by questions on the research topic, for example: “How did the training affect your professional views?”, “What specifically affected you and in what way?”, and “Was any part of the training less relevant?”.

#### 2.2.2. Written Assignment

During the second year, the course participants were provided with a written, summative assignment, part of which was used later for this study. In this part, the participants reflected on their personal and professional learning. It included questions such as: “Describe some factors that support or hinder therapeutic group work in educational settings”, and “In what ways was your work affected by this training?”. The assignments were submitted to the training coordinator anonymously, using the Google Forms interface, as a part of the course requirements.

### 2.3. Procedure and Ethics

During the first year, the 20 course participants were approached by the RSC office via email and provided with the contact info of an external, university research team. The participants’ employers were not involved in any other way in the recruitment or the research process. The participants were asked to sign informed consent forms, which noted that they could pull out of the study at any stage. Eleven out of the twenty course participants—four supervisors and seven creative arts therapists—agreed to take part in the study. Each participant was interviewed for 40–60 min, at the time and place of her choice; the interviews were recorded and transcribed with the participants’ consent. The recordings were deleted immediately after being transcribed and saved anonymously on a password-protected computer.

Since the second year, 34 written assignments were submitted anonymously; a university researcher reached out through the course WhatsApp group with the training coordinator’s permission and explained the second step in the study: accessing and analyzing their anonymous second-year assignments. Since no one objected, the written assignments, using the Google Forms interface, were accessed and analyzed. There were no dropouts at any stage. The study was approved by the Office of the Chief Scientist at the Israeli Ministry of Education (Permit No. 10471) and by the Ethics Committee of the Faculty of Social Welfare and Health Sciences at the University of Haifa (Permit No. 01/18 and 353/20).

### 2.4. Data Processing

The research method and the data analysis adhere to the principles of Consensual Qualitative Research [26,27], aiming to understand the subjective experience of the research participants. The initial step included three interviews and eight written assignments analyzed separately by three research team members; this preliminary analysis sought to identify and define domains. A group consultation followed, and the research team agreed on the domains as they emerged from the data. According to these domains, the rest of the interviews and the assignments were analyzed and divided into domains while constantly attending to additional, accumulating data. After that, each team member looked for core ideas emerging from every identified domain, and an additional consultation was held to define the core ideas. The third step, performed by a single researcher, entailed revisiting the raw data and analyzing them according to the domains and core ideas agreed upon during the consensus phase while indicating their frequency throughout the text. Two teams analyzed the accumulating data: each team included an experienced researcher and two art therapy students in the research track. An additional experienced researcher operated as an auditor at every step of the analysis.

The frequency of the appearance of core ideas within the interview data was categorized as follows: the phrase “most of the interviewees” indicates that the core idea came up in interviews of seven or more participants. “Some of the interviewees” represents a core idea that was mentioned by 3–6 participants, and “a small part of the interviewees” indicates a core idea that was mentioned by 2 participants or less. Regarding the written assignments, “the majority of the assignments” referred to the replies of from 21 to 34 creative arts therapists; “some of the assignments’’ referred to the replies of 10–20 creative arts therapists; “a minority of the assignments” referred to 3–9 replies, and “individual assignments” referred to 1–2 occurrences [26,27].

## 3. Results

### 3.1. The Experiences of Arts Therapists within the Education System

Most interviewees reported that their schools or kindergartens insisted on a greater extent of group work over recent years to let more children benefit from creative arts therapies. As a result, unique group formations were established, for example, pairs and trios: “They began seeing the students in pairs and calling them groups”. In addition, groups of up to 15 students, usually in special education classes, were facilitated based on a model called “The Class as a Group”, led by the therapist and the class teacher. Most interviewees preferred facilitating same-sex groups. For example, an interviewee said: “There is something in a group of boys and a group of girls, I think it is more convenient for me”.

#### 3.1.1. The Benefits of Group Arts Therapy in Educational Settings

Most interviewees said that group therapy in schools allowed children to practice their social skills within a peer group, experience diverse social roles, and feel “they were not alone”. Such experiences prepared them to try and interact socially outside of the group as well. Most interviewees commented that group therapy in school settings is more accessible to students: “It is easier for them to accept therapy that way; they do not feel like some aliens”. Most interviewees thought that group work in educational settings was important and suitable, as schools are basically social environments where the life span relates to the class groups or other interpersonal activities. According to some written assignments, the quality of therapy improved due to multi-professional collaboration at the school. One study participant wrote: “Several people work together: an educator, a school counselor, and a psychologist”. These staff members see the students daily and may offer different perspectives: “I can use the advice of quite a few staff members who sometimes witness the same group during different situations. Their view generates more knowledge on my group participants”.

#### 3.1.2. The Power of the Arts in Group Therapy

The study participants indicated in their written assignments that the artistic modalities (for example, music, visual arts, or drama) had special significance for their group work. The five main roles that the arts fulfill in group therapy were shared: (a) a natural means of expression and a processing resource for the children in therapy: “Through the arts, the children express personal issues, needs, desires, dreams, and fears, and in this way, they become more familiar with their creativity they can lead mental exploration and observation”; (b) the arts helped the group members to connect and socialize: “The arts enable more presence and connection, visibility. In a group of girls I have the artwork promotes amazing connections between girls who were more alienated and aggressive at the beginning of therapy”; (c) a projective mechanism: “In many sessions, we enter the dramatic play space, where each child picks a character, a role, and the whole group evolves out of the personal and interpersonal processes taking place within this intermediate space”; (d) a safe and inclusive space: “Since school is like a pressure cooker: demands, assignments, expectations from the students (and the staff as well), I feel that my therapy room is an island of sanity; a place where one can create with fewer expectations”; (e) a means for evaluation and assessment: “Through the arts they display another dimension hardly existing in verbal conversations; for example, during group play, you can learn a lot about the power relations: who leads, who stands aside, and more”.

#### 3.1.3. Group Facilitation in Educational Settings: An Intimidating Experience

Besides recognizing the advantages of group therapy in schools, most interviewees were discouraged and often tried to avoid facilitating groups, since they felt their skills were insufficient: “The schools’ demand to establish therapeutic groups is challenging. Facilitation skills are not taught adequately in our training programs, nor by the education system, so there is the problem”. Some therapists perceive their groups as an assortment of individuals, and that is why they see group therapy as a stressful task: “We think about the children like, for example, if you facilitate a group of four, and you see it them as four individual clients, it is not the right perspective”. This perspective led many creative art therapists in the education system to work with pairs of children defined as groups. Only a few experienced ones could share their choice to work with larger groups of children: “I’m used to working with groups of any possible size, and I have come to love it. It is not that usual for many other therapists in the education system”.

#### 3.1.4. Group Facilitation in Educational Settings: Disruptive Factors

Additionally, most interviews and written assignments described the arguable formation processes of their therapy groups, usually in partnership with the educational staff. According to the participants, a lack of understanding regarding the essence of therapeutic groups in the education system leads to group formations unsuited to children’s conditions and needs: “Very often, groups are composed due to the system’s needs and limitations and not the children’s needs or personal goals”. Most interviewees complained about the heavy workload they experienced regarding the number of students in the group and sometimes regarding several groups they facilitated. Additionally, there was not enough time to meet with parents: “It is different meeting two pairs of parents per a single therapeutic hour if you see a dyad in therapy or eight of them when you facilitate a group. This is something to be recognized and paid-for, and it isn’t!”. Most interviewees said it was difficult to find a suitable space at school to accommodate a group of more than three children. A few of the written assignments noted a shortage of resources: “Lack of equipment—creative art materials, board games, costumes, and accessories”. Most interviewees referred to the fixed schedules of schools, which sometimes clashed with the needs of the therapeutic group: “If I see over time that a child doesn’t fit into a certain group, to what extent can I make a change and see the child individually? During the school year I don’t have extra time slots”. Additionally, therapy is disrupted by religious holidays and summer vacations, and, according to the interviewees, it is hard to keep a regular and continuous process. Most interviewees and assignment writers claimed the time allotted for group therapy is insufficient: “45 min may be enough for a lesson in class, but not enough for an art therapy group”. Finally, some interviewees and a few of the assignment writers were concerned about the lack of confidentiality at schools since their group members met in other settings: “In class, during recess or in different lessons. This can ‘contaminate’ the therapeutic process”.

### 3.2. The Influences of the Group Facilitation Training

#### 3.2.1. Enhanced Confidence

Some interviewees said that, following the training, they could see the benefits of groups for the students in therapy. Considering their changing perceptions and the abilities they acquired during the training, most of the assignment respondents testified that their confidence as therapists and group facilitators improved: “This year, I feel more able working with groups; more relaxed and less anxious, more willing to observe the group and care for its needs”. Most interviewees noted that the training sparked their curiosity, thinking, and desire to gain more understanding of group therapy.

#### 3.2.2. Letting Go and Allowing the Group to Lead On

Most assignment respondents stated that the training enabled them to be less controlling and trust the group process: “I let things happen more freely, and I am here to mediate when required. I don’t structure or lead as much as I did before, some group processes happen naturally, and I must simply pay attention to them”. Following the training, most interviewees were more aware of the group’s ‘here and now’, watching the motion of content and dynamics minute-by-minute: “When I lead a group, I must be very attentive to the process and what is happening here and now”. Most participants learned not to avoid conflict or aggression in their groups and use it to promote the treatment process: “I am much less afraid of aggression within the group. It is a blessing for therapy”.

#### 3.2.3. The Importance of Group Work and the Perception of “the Group as a Whole”

Following training, some assignment respondents acknowledged the therapeutic importance of groups. They understood how group interactions were not interferences but learning opportunities for the children, who observed and interacted with each other and acquired more adaptive social behaviors. Most interviewees and some assignment respondents did not see the group as a collection of individuals anymore but as an entity. They realized the importance of finding the ‘group voice’, i.e., a theme or behavior expressed by an individual participant on behalf of the entire group. Capturing the ‘group voice’ assisted the entire group: “Understanding what a group voice is, that every individual statement is lived to some extent by every group participant and that I can point at every group member’s expression as if it represented the whole group”.

#### 3.2.4. Modifying the Group Formation Processes

Following training, some interviewees realized in more depth how group formation processes influenced the therapeutic outcome. Most interviewees, for instance, assembled larger groups. According to some, they understood that more participants in a group allow for additional voices to be heard and constructive processes to happen. Some interviewees noticed the contribution of diversity to their groups, and they experimented with putting together more heterogeneous groups: “I understand that it is all right to compose diverse groups, not only in terms of boys and girls but in all ways, for example, children on the autism spectrum with neurotypical kids”.

#### 3.2.5. Improved Group Facilitation Skills

An additional developing skill noted by some assignment respondents was an improved capacity to adjust their perspectives and intervention techniques to their groups’ nature and developmental stage: “Now I can comprehend the position of a group facilitator; I have a clearer understanding of the group process, the stages of development, I listen to the voices in a group and echo them more accurately”. Most interviewees described the developing communication with their groups as they learned how to listen to the ‘group voice’: “Attending to this voice is important. Not saying, for example, “you are telling us that it is difficult to be here”, but rather, “the group says it is difficult here now. It is essential to practice speaking this way”. Most interviewees shared that the training improved their ability to identify group processes. As a result, they reflected and echoed their groups’ ideas even more; they addressed the ‘here and now’ and the aggression in the room more boldly: “After training, I felt much more perceptive and could see what happened in each group at the present moment: the transference, the ‘here and now’, I could understand the girls in my group even better”. Finally, some interviewees learned to appreciate the importance of structuring the beginning and closure of their group sessions: “I never structured my sessions; my work has changed a bit, it’s something I do now even if resistance occurs. I say something, even a single sentence, like ‘how are you now?’ Something to begin with and to end the session”.

### 3.3. The Factors Enhancing the Learning Process

#### 3.3.1. Experiencing within a Psychodynamic Group

Most assignment writers believed that participating in an intimate, short-term psychodynamic group improved their learning process; being a part of a group helped them understand the processes that their therapy groups go through: “Participating in a group helped me experience my roles within a group and my personal development as related to the group’s developmental stages; I was able to transfer this knowledge and try to understand my therapy groups’ development”. Further to this, the psychodynamic group allowed the therapists to observe their facilitators at work: how they treated and managed them as a group or work out different situations that emerged.

#### 3.3.2. Learning Group Theory

Along with the shared group experience, some assignment respondents highlighted the importance of theoretical learning; it provided a broader perspective on the role of a group facilitator: “The lectures and workshops in various artistic modalities opened my mind to more ideas and other types of group facilitation”. Additionally, a minority of the creative arts therapists described that the theoretical learning improved their ability to observe their groups evolve: “My ability to identify the developmental stages of my groups improved, as did my professional understanding of the full therapeutic process”.

#### 3.3.3. The Benefits of Peer Learning

During the second year of training, the participants attended guest lectures delivered by some of their colleagues, who shared experiences and techniques from their group work in the education system. Most interviewees and a small part of the assignment respondents suggested that these sessions were very relevant to their everyday work experience with children: “These examples were helpful, diverse, and practical”.

### 3.4. Recommendations

#### 3.4.1. Recommendations for Group Therapy in the Education System

Some interviewees recommended working with up to four children in a group due to there not being appropriate spaces for accommodating more children and their art products, and the need to tune into the ‘here and now’ while managing the artwork and the children’s safety: “I’m an art therapist, and art is very present during our group sessions; larger groups are a matter of space, and also there is the issues of preserving the artwork, observing and not missing a detail”. Based on their new experiences, some interviewees thought that more heterogeneous groups based on children’s gender, age, and goals could enhance the therapeutic process. Furthermore, most interviewees suggested that groups with over four participants benefit from co-therapy: “Working in co-therapy allows a much better hold on the children”. Some added that processing time is essential in co-therapy. Most interviewees recommended that art group therapy sessions should last 90–120 min: “The 45-min-long session of my group did not suffice; 90 min are required when there are several children in a group, and you need time to process, and there is both conversation and something creative to do”. Some interviewees felt they needed a weekly supervision session pertaining to their group work: “Besides this course I receive no supervision on the subject of groups; it is something I need”.

#### 3.4.2. Recommendations for Group Facilitation Training to Creative Arts Therapists

Most interviewees emphasized the importance of undergoing more extensive group facilitation training during their basic creative arts therapies studies: “This is something that must be part of our training”. Most of the interviewees thought that the education system should offer therapists suitable group facilitation training: “When training programs do not provide it, the education system should as they enter the education system and are required to provide group therapy”. In addition, some interviewees believed that a professional facilitator must also experience group dynamics; after participating in group processes as a part of their training, the therapists’ ability to identify group processes improved: “I learned the most from my experiences as a group participant in this training, just as I did during group dynamics at the MA program”. According to the research participants, the experience of group dynamics in training should be complemented with relevant theory. Furthermore, some interviewees recommended placing an emphasis on the use of art materials and various other artistic techniques when working with groups of children and adolescents: “With larger groups at school now, I trust this form of therapy, but I feel the lack of intervention tools related to art materials and working with them or offering some free space for artwork”.

#### 3.4.3. Recommendations Regarding the Training

Most interviewees thought that the training was relevant, and they recommended that it continues annually. Some assignment respondents claimed that the theoretical learning was enjoyable and enriching; however, they stressed the need to improve it: “I would have preferred to gain a more profound theoretical understanding; we rushed through the theories and there was not enough time”. Some interviewees and a minority of the assignment respondents believed that the link between theory, field work, and the arts was not strong enough: “Our discourse often focused on adult groups rather than school children. The insights and the theories we learn should be based on situations that we encounter daily. The artistic modalities were not given equal attention to the psychodynamic learning”. Some interviewees and a minority of assignment respondents referred to the size of their training group and expressed a desire for smaller groups in the future; they argued that the group’s size made it harder to express themselves, participate, and ask for advice: “I would be very happy if there was a smaller group, we could then bring more cases for supervision”. A small number of the interviewees recommended adding more workshops to the curriculum to simulate the use of artistic media in groups.

## 4. Discussion

This study examined the perceptions of creative arts therapists and their clinical supervisors of group arts therapy in the Israeli education system, and the significance of the group facilitation training that they underwent. The interviewees described their experiences of facilitating group therapy in the education system. It seems that unique forms and models of group work have emerged in this field over the years: The “class as a group” model [28] and working with 2–3 students and defining them as “small groups” developed because of a demand to help more children, while taking into consideration that creative arts therapists find it difficult to manage larger groups, and many more clients every week. Sharim-Klein [29] claimed that, in this context, a pair is too small a unit as it does not allow children to benefit as they would within a larger peer group setting, with all the intensity, dynamics, resonance, and social feedback. This criticism is in line with the study participants’ views and indicates the need for the re-examination of this standard.

The research participants emphasized that group arts therapy in educational settings supported their clients. Many thought it allowed children to learn about themselves and communicate with their peers. In therapy groups, children show sympathy, forge partnerships, and their social confidence improves. These outcomes contribute to the social integration goal of the Israeli education system. They are also in keeping with the aims of group therapy in general and group arts therapy in particular [2,7,22]. The research participants felt that artistic media plays a significant role in their groups; some explained that the arts offered a non-verbal channel of communication. As the literature describes [4,5,6], the use of the arts helps children process emotional content and proactively explore their inner worlds. This outcome, despite the short periods of time provided for each session, supports other research findings on the contribution of the arts to group activities in school settings [19,20,30,31,32].

Additionally, the research touches on various problems such as those described in the literature pertaining to creative arts therapy in the Israeli education system, such as the excessive number of children in the groups and the lack of suitable time or spaces for this work [21,33]. In addition, the therapists felt they were insufficiently trained and, as a result, they were intimidated, burdened, and stressed by group work.

The findings also indicate that the training generated a perceptual change regarding group facilitation in the education system. It was not the first professional training offered to support a specific skill in the work of creative arts therapists in the education system; a similar outcome was achieved following a previous training course within the education system cultivating other skills [34]. The present study indicates participants’ improved attentiveness and heightened awareness as they facilitate their groups today. Notably, there is more interest and a desire to explore group work further. The research participants shared their increased confidence during group work. The old feeling that they are unable to monitor every child in their groups has changed. Through the “group as a whole” approach, which highlights the social context [24,25], some therapists could observe how their group behaves and communicates like a single living entity. The therapists also conveyed that now they were more aware of the ‘group voice’; in other words, they understood that an individual’s words or behavior could represent the whole group’s unaware processes [24]. Listening to the ‘group voice’, the therapists could better understand their groups.

The study participants described the benefits of learning through short-term psychodynamic groups; their first-hand learning process was portrayed as a profound developmental opportunity that broadened their understanding of group dynamics and the possible experiences of their group participants at school. Observing their facilitators during the course helped the creative arts therapists live the essence of group work, grasp the meaning of “the group as a whole”, and understand how they should allow their groups of children to interact more freely and independently. The psychodynamic group experience motivated them to seek new intervention methods for their clients at school. Yalom and Leszcz [1] emphasized the importance of self-experience during group facilitation training: in their view, practice groups function as microcosms allowing trainees to learn about themselves and other group members in a social context. The study participants suggested that, besides the psychodynamic group experience, future training should include more facilitation methods, models, and group experiences. They also noted the importance of dynamic group experiences or group facilitation simulations to acquire more skills and a broader perspective on group processes and dynamics. Indeed, various studies, such as [35], indicate that simulations during basic creative arts therapies training or later professional development facilitate reflective learning processes.

Additionally, the research shows a gap between the group arts therapies in schools and the theory and methodology offered in this training. They explained that the behaviors and needs of children at school were very different from the general group theory based on the research and experimentation with adult clients. Despite Yalom and Leszcz’s assertion [1] that the basic principles of group therapy apply to multiple populations and goals, they are less relevant to developmentally challenged or neurodivergent children, or to the special educational settings.

The interviewees claimed it was more complex to facilitate larger-size groups of children solely while observing the children’s behaviors, communications, and dynamics, facilitating conversation based on equality, and echoing the ‘here and now’. They suggested that co-therapy is more suitable for larger groups of children. Co-therapy, i.e., when two therapists facilitate a single group simultaneously, is well known in the world of group therapy; it is a parent-like model where two figures observe and manage the group, offering role models to its members regarding their cooperation, communication, and management. Co-therapy also offers group members more ways to interact and communicate with figures and topics concerning authority, power, and control. Co-therapy may be hard to implement in general due to budgetary or staff restrictions, but, even when the interviewees found their partners at school, it was hard to find themselves processing time. It is important to mention that co-therapy requires both facilitators to maintain an ongoing and open dialogue [36]; therefore, regular processing time is essential: it allows for time to reflect on the mutual dialogue and the group’s development [37].

During the two-year training, the participants noted the contribution of group-adapted supervision to their daily work in the education system. One affordable idea that might assist less-experienced therapists that still are not trained in group work is mentoring: co-facilitating groups of children with another who is more experienced with groups [38]; this type of mentoring may offer opportunities for learning [36,39,40] by observing the other therapist in action while experiencing the group process firsthand. Shargai and Hoffman [41] add that the co-facilitation of groups improves therapists’ clinical skills and knowledge and increases self-awareness and self-confidence.

## 5. Limitations and Future Research

The present study has several limitations: firstly, the participants were only a small percentage of over 3600 creative arts therapists who work in the Israeli education system. They are all employed in a single region in the center of Israel, and they may or may not represent the experiences of other therapists in other locations. In addition, this qualitative study focused on the subjective experiences of therapists and changes that followed their group facilitation training. A follow-up study could examine the continued impact of the training in other aspects such as the efficacy of therapy, these therapists’ perceptions of group arts therapy over time and as they gain more experience, and the wider impact of such courses within the education system.

## Data Availability

The datasets for this manuscript are not publicly available to protect the participants’ confidentiality. Requests to access the datasets should be directed to E.R.

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
