# Peer review of "The Meaning of a Group Facilitation Training to Creative Arts Therapists Working in the Public Education System"

_children, 2023, doi:10.3390/children10060933_

Round 1

Reviewer 1 Report

Dear Authors,

Congratulations on writing such a clear and informative article, which is very well-written and comprehensive. I have a few minor corrections and suggestions for your consider:

Abstract

- A note on the timeframe of the studies would be useful. Were they part of one project, with two stages? Or two separate projects that combined unexpectedly?

Introduction

- Are art therapies more common in Israeli private schools? Some contextualisation of the Israeli educational system would be helpful, e.g., in terms of class sizes and budgetary challenges.

- Avoid contractions ("it's") in academic English, unless this is within the participants' natural speech.

Methods

- What were the genders of the participants? You mention "her" for all the participants later on.

- Some references supporting the strength / relevance of the selected methods would be helpful.

- Ethics: Did any of the participants withdraw? How was the data handled?

Results

- Some links between the subsections would help with the flow of the article.

- Line 186: Extra space added?

- Line 197: Through or trough?

- Line 221: Was not (only use contractions within the data, as in the way people would speak in English).

Discussion

- Line 393: We use "the children" for a specific group of children in question, so when you refer to the wider literature, please just use "children."

Conclusions

- What percentage of the 3600 creative arts therapists in Israel were in your sample, exactly?

- Final words seem to be in a different font size (as in the Author Contributions section).

Overall, the English language level is very high. Some reference to the translation from Hebrew to English would be helpful for future researchers trying such a study outside the Anglosphere.

Reviewer 2 Report

The review results are as follows.

1.       Present your findings and conclusions in the abstract.

2.       Explain the hypothesis of this study in a schematic way.

3.       Describe the recruitment process, including the recruitment method of the study participants.

4.       Describe the criteria for selecting and excluding research participants.

5.       Describe the enroll process of the annual target participiants.

6.       Describe the time, place, and environment of the interview.

7.       Describe in numbers when the total study period, the first period, and the second period mean. If this study was conducted during the COVID-19 pandemic, quarantine rules should be described.

8.       Describe the difference in the type and number of participants between the first and second years.

9.       Describe the dropout rate of research participants and why.

10.    Present the demographic and sociological characteristics of the participants as a table.

11.    Explain the response analysis and the process of deriving results of this qualitative study using a schematic.

Reviewer 3 Report

The study employed a qualitative study design with inherent subjectivity. However, the authors have made a good attempt to study the important topic and make useful recommendations that are practical in nature to enhance the effectiveness of creative art therapy outcomes for school children.

There is a need to address some language editing and spelling mistakes.  

Round 2

Reviewer 2 Report

This manuscript was appropriately revised based on the reviewer's comments.

Thank you for your effort.